# The Role of Gender in the Association between Mental Health and Potentially Preventable Hospitalizations: A Single-Center Retrospective Observational Study

**DOI:** 10.3390/ijerph192214691

**Published:** 2022-11-09

**Authors:** Fabrizio Cedrone, Alessandro Catalini, Lorenzo Stacchini, Nausicaa Berselli, Marta Caminiti, Clara Mazza, Claudia Cosma, Giuseppa Minutolo, Giuseppe Di Martino

**Affiliations:** 1Health Management of “S. Spirito” Hospital, Local Health Authority of Pescara, 65100 Pescara, Italy; 2Department of Medicine and Ageing Sciences, “G. d’Annunzio” University of Chieti-Pescara, 66100 Chieti, Italy; 3Department of Biomedical Sciences and Public Health, Università Politecnica delle Marche, 60100 Ancona, Italy; 4Department of Health Science, University of Florence, 50100 Florence, Italy; 5Department of Biomedical, Metabolic and Neurosciences, University of Modena and Reggio Emilia, 41100 Modena, Italy; 6Department of Medicine and Surgery—Sector of Public Health, University of Perugia, 06100 Perugia, Italy; 7Department of Public Health, Experimental and Forensic Medicine, University of Pavia, 27100 Pavia, Italy; 8Department of Health Promotion, Mother and Child Care, Internal Medicine and Medical Specialties, University of Palermo, 90100 Palermo, Italy; 9Unit of Hygiene, Epidemiology and Public Health, Local Health Authority of Pescara, 65100 Pescara, Italy

**Keywords:** mental health, substance abuse, alcohol abuse, barriers, gender inequalities, preventable hospitalizations, public health

## Abstract

Psychiatric disorders and substance abuse are barriers that limit access to timely treatment and can lead to Potentially Preventable Hospitalizations (PPH). The aim of this study is to identify the role played by gender in the association between mental health and PPH. Hospital discharge records (HDRs) from the Local Health Authority of Pescara (Italy) from 2015 to 2021 were examined and PPH were measured according to Prevention Quality Indicators (PQIs) provided by the Agency for Healthcare Research and Quality. In total, 119,730 HDRs were eligible for the study and 21,217 patients fell into the PQI categories. Mental health disorders and addictions were extracted from the HDRs through the Elixhauser Enhanced ICD-9-CM algorithm. The association between PQI hospitalization and some predictors considered was evaluated with multivariate logistic regression models. In males and females, alcohol abuse showed a protective role towards preventable admissions for PQI-90 (all types of conditions) and PQI-92 (chronic conditions). In contrast, there is a gender gap in accessibility to primary health care, especially for acute conditions leading to PPH. Indeed, in males, PQI-91 admissions for acute conditions were found to be positively associated with drug abuse, psychosis, and depression, whereas this was not the case for females.

## 1. Introduction

Potentially Preventable Hospitalizations (PPH) for physical health conditions are used as a measure of health service access, integration, and effectiveness. PPH are specific hospital admissions that could potentially have been avoided through preventative health interventions and community care: long term studies found PPH elevated in patients with mental health conditions [1].

Barriers to primary care access lead to PPH. These barriers can be driven by psychological factors that can make it difficult for patients to prioritize their own health; in addition, primary care providers’ stigma on mental health and substance use can have an impact on the extent to which patients communicate their needs [2]. Analyzing PPH can indirectly help to define the determinants of the barriers to primary care access.

Gender roles can have an impact on generating differences in prevalence rates and determinants of mental health indicators between women and men [3]. In fact, the gender gap in mental health service use is caused not only by men’s negative attitudes toward help seeking, but also by structured social norms associated with masculinity [4]. For example, men are more likely to report barriers to care related to their perceptions of mental health issues and usefulness of health care services and are less likely to acknowledge the helpfulness of psychotherapy [4,5].

Using appropriate, inclusive, interprofessional models for mental health care is fundamental to enhance the delivery of mental health care in primary care settings and improve health outcomes for people with mental disorders [6,7,8]. In order to choose the proper model, to evaluate its efficacy and accessibility for psychiatric patients, local data inflow such as hospital discharge records can become an asset. Hospital discharge records (HDRs) allow estimates of the cost of hospitalizations that can be an indicator of the efficiency of primary health care: it can be used to identify PPH and orient programmatic decisions, both in hospital and primary care settings [9,10]. Prevention Quality Indicators (PQIs) created by the Agency for Healthcare Research and Quality (AHRQ) use data from HDRs to identify hospitalizations that could be preventable with access to quality primary health care and identify barriers to primary care access. PQIs result as a key tool to evaluate community health needs and primary health care services [11].

Current research on PPH still lacks focus on gender aspects, especially the one regarding mental health. The aim of the study is to identify gender disparities in the association between PPH and patients’ mental health state.

## 2. Materials and Methods

We carried out a retrospective observational study by analyzing the admissions that occurred in the Local Health Authority (LHA) of Pescara, the most populated city in Abruzzo, a region of Southern Italy. The LHA of Pescara serves a catchment area of about 320,000 inhabitants and is organized with a large hub hospital and two spoke hospitals [12]. Data were collected from hospital discharge records (HDRs) of all hospital admissions that occurred from 1 January 2015 to 31 December 2021. All the patients living in the LHA of Pescara were selected. The HDRs collected information on admission source and type, admission and discharge dates, patient baseline demographics (age, gender, citizenship, birthplace, and residence), the principal diagnosis and up to five additional diagnoses, the main procedure and up to five further procedures performed during the admission. Diagnoses and procedures were coded according to the International Classification of Diseases, 9th Revision, Clinical Modification (ICD-9-CM, National Center for Health Statistics (NCHS), and the Centers for Medicare and Medicaid Services External, Atlanta, GE, USA). To compute the preventable admission rate, AHRQ definitions [13] were followed, including four different Prevention Quality Indicators: Prevention Quality Overall Composite (PQI-90), Prevention Quality Acute Composite (PQI-91), Prevention Quality Chronic Composite (PQI-92), Prevention Quality Diabetes Composite (PQI-93). Appendix A provides details on the definition, use, and composition of PQI Composite Measures (Table A1). Physical comorbidities and those specifically affecting mental health (depression and psychosis) and addictions (alcohol and drugs) were extracted from HDRs through the Elixhauser enhanced ICD-9-CM algorithm proposed by Quan et al. [14].

### Statistical Analysis

Categorical variables were reported as frequency and percentage. Pearson’s Chi-square test was performed to compare baseline variables between gender for each of the PQIs examined. To evaluate the statistical association between PQI hospitalization and the various predictors taken into consideration, multivariate logistic regression models were implemented, one for the whole sample and one for the male and female sample. Statistical significance was set at *p*-value < 0.05. All analyses were performed with Stata^®^ version 15 (StataCorp LLC, College Station, TX, USA).

## 3. Results

As showed in Figure 1, of the 252,755 hospital discharge records produced by the three hospitals in the Pescara LHA, 95,558 are excluded because residents out of the Pescara LHA. In total, 23,300 HDRs belonging to patients under the age of 18 and 13,253 HDRs related to trauma are excluded. Finally, 934 HDRs are excluded for other reasons such as coding errors. In total, 119,730 are eligible for our study and 21,217 patients fall into the PQIs categories.

The characteristics of the sample are summarized in Table 1. About the study population, 47.07% of included patients were males. The majority of patients are between 61 and 80 years of age (36.82%), followed by patients aged over 80 (22.26%), 41–60 (21.67%), and 18–40 (19.25%). In total, 50.27% of patients have one or more comorbidities.

In the overall sample, patients who have psychiatric or addiction problems under study are 3979 (3.32%). Of these, 2557 have at least one psychiatric diagnosis (2.14%), 1387 have at least one addiction diagnosis (1.16%), and 142 have both diagnoses (0.12%).

The hospital discharge records that fall into the PQIs categories are 21,217, of which 11,547 are about men (54.42%) and 9670 are about women (45.58%). Age of the population by gender and PQI are shown in Table 2. The majority of patients are between 61 and 80 years of age (9322, 43.94%), followed by the categories of over 80 (8884, 41.87%), 41–60 (2384, 11.24%), and 18–40 (627, 2.95%) years old. There are statistically significant differences between males and females by age: considering the PQI-90 overall and the PQI-91 acute hospitalizations, females are predominant in the 18–40 and over 80 age groups, while they are less represented in the 41–60 and 61–80 age groups. Females are always underrepresented in the PQI-93 diabetes hospitalization and overrepresented only in the over 80 age group considering the PQI-92 chronic hospitalization.

Characteristics (physical comorbidities and mental health/substance use conditions) of the population by gender and PQI are shown in Table 3. In total, 90.61% (19,224) hospital discharge records considered in PQIs concern patients with one or more comorbidities and most have three or more comorbidities (38.90%).

Considering all the hospital discharge records that fall into the PQIs categories, 53 (0.25%) of them concern drug-using patients, 109 (0.51%) alcohol users, 170 depressed patients, and 119 (0.80%) subjects with psychosis, 314 (1.48%) concern at least one psychiatric diagnosis, 162 (0.76%) at least one addiction diagnosis, and 4 (0.02%) both diagnoses.

In males, depression and having at least one psychiatric diagnosis were risk factors for PQI-90 overall hospitalization (OR = 1.69, 95% CI: 1.19–2.38 and OR = 1.37, 95% CI: 1.05–1.76). Drug abuse, psychosis, depression, having at least one psychiatric diagnosis, and having at least one addiction diagnosis were risk factors for PQI-91 acute hospitalization (OR = 2.02, 95% CI: 1.23–3.28, OR = 1.59, 95% CI: 1.02–2.45, OR = 2.44, 95% CI: 1.56–3.81, OR = 2.13, 95% CI: 1.53–2.93, OR = 1.64, 95% CI: 1.15–2.32, respectively). Data analysis also shows a growing positive association between the age of the patients and the increase of comorbidities. All the results are shown in Table 4.

The categories “Alcohol abuse” and “At least one addiction diagnosis” seem to prevent PQIs hospitalization except for the PQI-91 acute one (OR = 0.43, 95% CI: 0.31–0.58 and OR = 0.57, 95% CI: 0.43–0.73, respectively, for the association of alcohol abuse and at least one addiction diagnosis with PQI-90 overall; OR = 0.33, 95% CI: 0.22–0.48 and OR = 0.34, 95% CI: 0.23–0.48, respectively, for the association of alcohol abuse and at least one addiction diagnosis with PQI-92 chronic; OR = 0.32, 95% CI: 0.15–0.68 and OR = 0.43, 95% CI: 0.22–0.80, respectively, for the association of alcohol abuse and at least one addiction diagnosis with PQI-93 diabetes).

In females (Table 5), the categories “Alcohol abuse” and “At least one addiction diagnosis” seem to prevent PQI-90 overall and PQI-92 chronic hospitalizations (OR = 0.39, 95% CI: 0.15–0.99 and OR = 0.44, 95% CI: 0.20–0.96, respectively, for the association of alcohol abuse and at least one addiction diagnosis with PQI-90 overall; OR = 0.19. 95% CI: 0.04–0.77 and OR = 0.16, 95% CI: 0.03–0.65, respectively, for the association of alcohol abuse and at least one addiction diagnosis with PQI-92 chronic). There is no evidence of risk factors for psychosis, depression, drug abuse, and addiction diagnosis. As for males, data analysis shows a growing positive association between the age of the patients and the increase in comorbidities.

## 4. Discussion

Our results show that, in males, there is a clear positive association between the four investigated psychiatric conditions/addictions and PQI-91 acute potentially preventable admissions, with the exception of alcohol abuse for which the association is not statistically significant. On the contrary, in females, having a mental health condition/addiction does not seem to increase the risk of undergoing an avoidable acute hospitalization.

Research has consistently found that men of different ages, nationalities, and social backgrounds are less likely than women to seek professional help for physical and mental health problems [15,16,17]. This means that men are more at risk of developing complications from their medical conditions leading to avoidable hospitalizations, as evidenced by our results for acute preventable hospitalizations. The social construction of masculinity, which requires a man to provide for his problems with his strengths, and the stigma of mental health conditions, with its feeling of shame and blame, are known to be some of the determinants that most affect men’s healthcare service use [4,18]. Furthermore, women tend to use preventive and diagnostic services more frequently, whereas men make greater use of emergency services [19,20]. It is assumed that these determinants could play a role even in our population. However, specific research about the use of non-psychiatric healthcare services by patients with psychiatric/addiction problems is needed to confirm this.

PQI-91 is a composite indicator that includes acute preventable admissions for urinary tract infections and bacterial pneumonia. Treating these diseases promptly when the first symptoms appear is certainly a primary goal. However, the prevention of these conditions is a more ambitious health goal even if not more difficult to achieve. Considering that streptococcus pneumoniae is the pathogen most responsible for bacterial pneumoniae, a strong campaign to promote pneumococcal vaccination, aimed specifically at male patients with these mental health conditions, is one of the possible ways to reduce acute avoidable hospitalizations in our Local Health Authority [21].

In males, there are no other significant associations of mental illness neither with preventable hospitalizations for chronic conditions nor specifically for hospitalizations related to diabetes. Chronic conditions such as those included in PQI-92 and PQI-93 (diabetes, COPD, asthma, hypertension, and heart failure) offer a longer period between the onset of symptoms and the occurrence of complications worthy of hospitalization [12]. This means that there is more time for the general practitioner or for other specialists to intercept the disease and take care of the patient, even with mental health comorbidities. Otherwise, acute conditions, included in PQI-91 composite indicator, such as bacterial pneumonia and urinary tract infections tend to evolve quickly, leading in a short time to complications and hospitalizations. One hypothesis to explain our findings is that males with mental health conditions face barriers that prevent access to general medical care in a short time.

Many general practitioners in Italy, even before the COVID-19 pandemic, had abandoned the organization of the open-access clinic in favor of the organization of an appointment diary [22]. This is one of the reasons that could explain why accessing care for acute problems could be so challenging for this group of patients. Making and meeting an appointment is not an easy task for a patient with mental health problems or addictions, because of the mental disease symptoms but also for the socioeconomic factors that usually match these conditions [2,23]. Creating specific agendas for access to medical care for this fragile population group is one of the possible solutions.

The protective role of alcohol abuse (and more generally of the category “At least one diagnosis of addiction”) towards PQI-90 overall and PQI-92 chronic preventable hospitalizations in females is peculiar [23]. However, this result could be influenced by the low number of women with addictions included in PQI groups. On the contrary, in men, whose numerical representativeness is greater in PQI groups, the protective role of alcohol abuse is even stronger and extends to PQI-93 diabetes-related preventable admissions [12,24].

Studies have showed that certain mental health conditions lead the patient to refer to the general practitioner more often. This is the case of generalized anxiety disorder, post-traumatic stress disorder, and panic disorder [25]. However, when focusing on alcohol abuse, many studies highlight how this condition is linked with fewer primary health care visits and more non-psychiatric hospitalizations [26]. Further investigations should be carried out to understand these results, which could arise from study limits but also, for example, from a particular and as yet undefined characteristic of the population under study or from a good functioning of the territorial services for alcohol addiction.

In our study, we performed a logistic regression to assess the association of mental conditions to preventable association correcting for age and physical comorbidities. Further elements, such as socioeconomic condition and consumption of medications, may be considered to further correct our findings for a more in-depth study.

The overall number of patients in the PQI categories represents a limit to the understanding of the phenomenon, especially in females for whom drug and alcohol abuse are less frequent even if in sharp growth [27,28]. Extending the observation period of our study can bring out associations that cannot be highlighted with the current sample. However, the overall number of patients in PQI groups is also linked to the coding of the hospital discharge records (HDRs). Although the coding of HDRs has been regulated by ministerial guidelines and the reproducibility has significantly improved over the years, there is still a fair amount of variability among healthcare professionals in the choice of the ICD-9-CM codes to be identified as the main and secondary diagnosis [29]. The main diagnosis code is fundamental, for our study and according to the indications of the AHRQ, to identify or exclude a preventable hospitalization. The secondary diagnoses codes, on the other hand, are equally important to identify patients with mental health conditions or addiction. It is documented that in Italy, mental health conditions are clearly underdiagnosed and, therefore, not coded (especially when it comes to comorbidities) among the hospital discharge record diagnoses.

The AHRQ definitions are one of the possibilities to identify potentially preventable hospitalizations. Although well-validated and provided with exclusion criteria for comorbidities that would make them not preventable, there could be some hospitalizations identified that are not preventable. Conversely, hospitalizations for other diagnoses not identified by AHRQ criteria may have been potentially preventable.

Finally, our data concern a limited population of a Local Health Authority and may not be generalizable.

## 5. Conclusions

Gender can have an impact on the accessibility of primary care for patients with psychiatric pathologies or addictions. Quality prevention for patients with mental health conditions or addictions and improved access to psychiatric and non-psychiatric services may limit potentially preventable hospitalizations.

The gender lens we applied to our population may be a successful approach to health inequalities: women and men respond to different diagnostic-prescriptive appropriateness, which depends both on biology and on social, cultural, psychological, and economic differences.

In a public health system where equity of access to care is one of the dimensions of quality, research in this area can help identify the most vulnerable population subgroups and guide targeted health policies.

## Figures and Tables

**Figure 1 ijerph-19-14691-f001:**
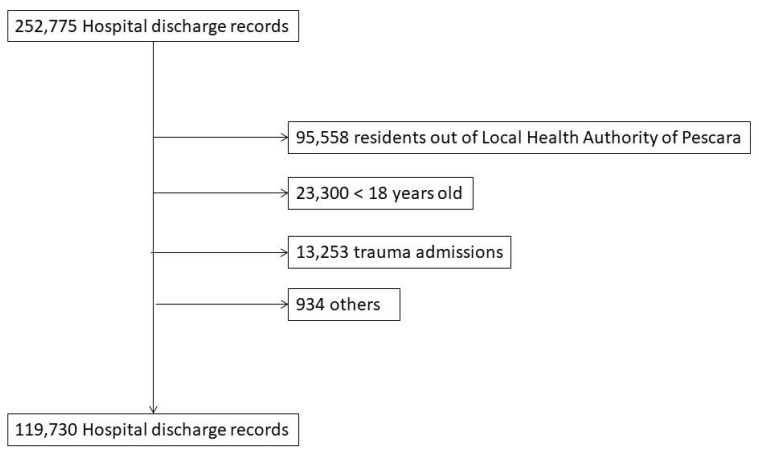
Flow chart of included HDRs.

**Table 1 ijerph-19-14691-t001:** Sex, age category, and physical comorbidities of the population under study and of the subjects considered in PQIs.

N (%)	Total Population	PQI-90 Overall	PQI-91 Acute	PQI-92 Chronic	PQI-93 Diabetes
	119,730	9902	2429	7492	1394
Sex					
Males	56,362 (47.07)	5313 (53.66)	1277 (52.57)	4047 (54.02)	910 (65.28)
Age category					
18–40	23,045 (19.25)	304 (3.07)	220 (9.06)	84 (1.12)	19 (1.36)
41–60	25,945 (21.67)	1106 (11.17)	412 (16.96)	696 (9.29)	170 (12.20)
61–80	44,089 (36.82)	4234 (42.76)	917 (37.75)	3330 (44.45)	841 (60.33)
Over80	26,651 (22.26)	4258 (43.00)	880 (36.23)	3382 (45.14)	364 (26.11)
Physical comorbidities (Num)					
0	59,545 (49.73)	993 (10.03)	857 (35.28)	137 (1.83)	6 (0.43)
1	28,533 (23.83)	2152 (21.73)	725 (29.85)	1430 (19.09)	218 (15.64)
2	19,594 (16.37)	2977 (30.06)	586 (24.13)	2401 (32.05)	482 (34.58)
3+	12,058 (10.07)	3780 (38.17)	261 (10.75)	3524 (47.04)	688 (49.35)

3+: more than three comorbidities.

**Table 2 ijerph-19-14691-t002:** Age of the population by gender and by PQI.

	PQI-90 Overall			PQI-91 Acute			PQI-92 Chronic			PQI-93 Diabetes		
	Males	Females	*p* Value	Males	Females	*p* Value	Males	Females	*p* Value	Males	Females	*p* Value
Total	5313	4589		1277	1152		4047	3445		910	484	
Age												
18–40	135 (2.54)	169 (3.68)	<0.001	79 (6.19)	141 (12.24)	<0.001	56 (1.38)	28 (0.81)	<0.001	11 (1.21)	8 (1.65)	<0.001 **
41–60	737 (13.87)	369 (8.04)	<0.001	238 (18.64)	174 (15.10)	<0.001	501 (12.30)	195 (5.66)	<0.001	130 (14.29)	40 (8.26)	<0.001 **
61–80	2581 (48.58)	1653 (36.02)	<0.001	537 (42.05)	380 (32.99)	<0.001	2051 (50.68)	1279 (37.13)	<0.001	570 (62.64)	271 (55.99)	<0.001 **
Over 80	1860 (35.01)	2398 (52.26)	<0.001	423 (33.12)	457 (39.67)	<0.001	1439 (35.56)	1943 (56.40)	<0.001	199 (21.87)	165 (34.09)	<0.001 **

The use of double asterisk (**) indicates that statistical significance was tested with Fisher’s exact test, since the sample size was small.

**Table 3 ijerph-19-14691-t003:** Characteristics (physical comorbidities and mental health/substance use conditions) of the population by gender and by PQI.

	PQI-90 Overall			PQI-91 Acute			PQI-92 Chronic			PQI-93 Diabetes		
	Males	Females	*p* Value	Males	Females	*p* Value	Males	Females	*p* Value	Males	Females	*p* Value
Total	5313	4589		1277	1152		4047	3445		910	484	
Physical comorbidities (Num)												
0	496 (9.34)	497 (10.83)	0.056	408 (31.95)	449 (38.98)	0003	89 (2.20)	48 (1.39)	0.040	5 (0.55)	1 (0.21)	0.030 **
1	1172 (22.06)	980 (21.36)	0.056	393 (30.78)	332 (28.82)	0003	782 (19.32)	648 (18.81)	0.040	127 (13.96)	91 (18.80)	0.030 **
2	1629 (30.66)	1348 (29.37)	0.056	327 (25.61)	259 (22.48)	0003	1306 (32.27)	1095 (31.79)	0.040	308 (33.85)	174 (35.95)	0.030 **
3+	2016 (37.94)	1764 (38.44)	0.056	149 (11.67)	112 (9.72)	0003	1870 (46.21)	1654 (48.01)	0.040	470 (51.65)	218 (45.04)	0.030 **
Drug abuse	23 (0.43)	2 (0.04)	<0.001 **	18 (1.41)	2 (0.17)	0.001 **	5 (0.12)	0 (0.00)	0.046 **	3 (0.33)	0 (0.00)	0.206 **
Alcohol Abuse	45 (0.85)	5 (0.11)	<0.001 **	17 (1.33)	3 (0.26)	0.003 **	29 (0.72)	2 (0.06)	<0.001 **	7 (0.77)	1 (0.21)	0.275 **
Depression	40 (0.75)	56 (1.22)	0018	21 (1.64)	17 (1.48)	0738	29 (0.72)	2 (0.06)	<0.001 **	2 (0.22)	3 (0.62)	0.349
Psychosis	34 (0.64)	21 (0.46)	0.223	22 (1.72)	7 (0.61)	0.014 **	12 (0.30)	14 (0.41)	0420	5 (0.55)	4 (0.83)	0.508 **
At least one psychiatric diagnosis	73 (1.37)	77 (1.68)	0.217	42 (3.29)	24 (2.08)	0080	31 (0.77)	53 (1.54)	0001	7 (0.77)	7 (1.45)	0.263
At least one addiction diagnosis	68 (1.28)	7 (0.15)	<0.001 **	35 (2.74)	5 (0.43)	<0.001	34 (0.84)	2 (0.06)	<0.001 **	10 (1.10)	1 (0.21)	0.110
Both diagnoses	2 (0.04)	0 (0.00)	0.503 **	1 (0.08)	0 (0.00)	1.000 **	1 (0.02)	0 (0.00)	1.000 **	0 (0.00)	0 (0.00)	-

The use of double asterisk (**) indicates that statistical significance was tested with Fisher’s exact test, since the sample size was small; 3+: more than three comorbidities

**Table 4 ijerph-19-14691-t004:** Logistic regression for PQI in males.

	PQI-90 Overall			PQI-91 Acute			PQI-92 Chronic			PQI-93 Diabetes		
	Odds Ratio (95%)		*p* Value	Odds Ratio (95%)		*p* Value	Odds Ratio (95%)		*p* Value	Odds Ratio (95%)		*p* Value
Age												
18–40	Ref.	-	-	Ref.	-	-	Ref.	-	-	Ref.	-	-
41–60	1.25	(1.02–1.5)	0.025	1.14	(0.87–1.47)	0.324	1.41	(1.05–1.87)	0.019 *	1.50	(0.8–2.8)	0.203
61–80	1.46	(1.21–1.75)	<0.001 *	1.28	(0.99–1.64)	0.050	1.65	(1.24–2.17)	<0.001 *	1.74	(0.94–3.18)	0.076
over 80	2.05	(1.69–2.46)	<0.001 *	2.15	(1.65–2.77)	<0.001 *	2.12	(1.6–2.8)	<0.001 *	1.01	(0.54–1.87)	0.971
Physical comorbidities (Num)												
0	Ref.	-	-	Ref.	-	-	Ref.	-	-	Ref.	-	-
1	3.54	(3.17–3.95)	<0.001 *	1.37	(1.18–1.58)	<0.001 *	12.91	(10.33–16.11)	<0.001 *	38.86	(15.86–95.21)	<0.001 *
2	7.45	(6.69–8.28)	<0.001 *	1.60	(1.36–1.86)	<0.001 *	32.15	(25.83–40.01)	<0.001 *	135.00	(55.57–327.92)	<0.001 *
3+	17.39	(15.61–19.37)	<0.001 *	1.10	(0.9–1.34)	0.326	87.39	(70.24–108.71)	<0.001 *	355.34	(146.46–862.1)	<0.001 *
Alcohol abuse	0.43	(0.31–0.58)	<0.001 *	1.05	(0.64–1.7)	0.853	0.33	(0.22–0.48)	<0.001 *	0.32	(0.15–0.68)	0.003 *
Drug abuse	1.17	(0.75–1.79)	0.490	2.02	(1.23–3.28)	0.005 *	0.42	(0.17–1.02)	0.056	1.22	(0.38–3.9)	0.734
Psychoses	1.08	(0.75–1.54)	0.686	1.59	(1.02–2.45)	0.038 *	0.62	(0.34–1.13)	0.121	1.14	(0.46–2.81)	0.777
Depression	1.69	(1.19–2.38)	0.003 *	2.44	(1.56–3.81)	<0.001 *	1.14	(0.69–1.86)	0.605	0.49	(0.12–1.99)	0.320
At least one psychiatric diagnosis	1.37	(1.05–1.76)	0.017 *	2.13	(1.53–2.93)	<0.001 *	0.85	(0.58–1.25)	0.425	0.85	(0.39–1.82)	0.683
At least one addiction diagnosis	0.57	(0.43–0.73)	<0.001 *	1.64	(1.15–2.32)	0.006 *	0.34	(0.23–0.48)	<0.001 *	0.43	(0.22–0.8)	0.008 *
Both diagnoses	0.64	(0.14–2.79)	0.555	0.17	(0.02–1.29)	0.088	1.52	(0.18–1.22)	0.695	1.00	(0–0)	

The use of asterisk (*) indicates the statistical significance. 3+: more than three comorbidities

**Table 5 ijerph-19-14691-t005:** Logistic regression for PQI in females.

	PQI-90 Overall			PQI-91 Acute			PQI-92 Chronic			PQI-93 Diabetes		
	Odds Ratio (95%)		*p* Value	Odds Ratio (95%)		*p* Value	Odds Ratio (95%)		*p* Value	Odds Ratio (95%)		*p* Value
Age												
18–40	Ref.	-	-	Ref.	-	-	Ref.	-	-	Ref.	-	-
41–60	1.45	(1.19–1.76)	<0.001 *	1.55	(1.23–1.94)	<0.001 *	1.78	(1.18–2.69)	0.006 *	0.89	(0.4–1.91)	0.756
61–80	2.56	(2.14–3.06)	<0.001 *	2.24	(1.81–2.77)	<0.001 *	3.60	(2.42–5.34)	<0.001 *	1.71	(0.83–3.52)	0.142
Over 80	4.23	(3.53–5.05)	<0.001 *	3.39	(2.73–4.2)	<0.001 *	5.87	(3.96–8.7)	<0.001 *	0.99	(0.47–2.05)	0.979
Physical comorbidities (Num)												
0	Ref.	-	-	Ref.	-	-	Ref.	-	-	Ref.	-	-
1	3.57	(3.17–4.02)	<0.001 *	1.36	(1.16–1.59)	<0.001 *	22.17	(16.34–30.06)	<0.001 *	221.90	(30.34–1622.45)	<0.001 *
2	7.52	(6.68–8.45)	<0.001 *	1.50	(1.26–1.78)	<0.001 *	56.54	(41.78–76.5)	<0.001 *	630.16	(86.34–4598.79)	<0.001 *
3+	18.89	(16.77–21.27)	<0.001 *	0.99	(0.78–1.23)	0.901	163.63	(120.93–221.39)	<0.001 *	1326.64	(181.6–9691.2)	<0.001 *
Alcohol abuse	0.39	(0.15–0.99)	0.048 *	1.40	(0.44–4.44)	0.565	0.19	(0.04–0.77)	0.021 *	0.61	(0.08–4.39)	0.622
Drug abuse	0.47	(0.11–1.95)	0.303	1.14	(0.28–4.64)	0.853	1.00	(0–0)		1.00	(0–0)	
Psychoses	0.87	(0.54–1.37)	0.541	0.87	(0.41–1.84)	0.720	0.85	(0.47–1.49)	0.566	1.43	(0.52–3.91)	0.485
Depression	1.03	(0.77–1.38)	0.825	1.24	(0.76–2.01)	0.392	0.94	(0.65–1.33)	0.716	0.45	(0.14–1.4)	0.169
At least one psychiatric diagnosis	0.99	(0.77–1.26)	0.933	1.13	(0.75–1.7)	0.553	0.91	(0.67–1.23)	0.556	0.74	(0.34–1.58)	0.443
At least one addiction diagnosis	0.44	(0.2–0.96)	0.040 *	1.46	(0.59–3.57)	0.405	0.16	(0.03–0.65)	0.011 *	0.49	(0.06–3.54)	0.482
Both diagnoses	1.00			1.00	(0–0)		1.00	(0–0)		1.00	(0–0)	

The use of asterisk (*) indicates the statistical significance. 3+: more than three comorbidities

## Data Availability

Authors can be contacted for information about dataset.

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
