# Peer review of "The Role of Gender in the Association between Mental Health and Potentially Preventable Hospitalizations: A Single-Center Retrospective Observational Study"

_ijerph, 2022, doi:10.3390/ijerph192214691_

Round 1

Reviewer 1 Report

Interesting topic. few clarifications which is mentioned in the pdf.

 FOR EDITORS

  1. The manuscript is relevant to the field and it is structured systematically.
  2. Most of the references are recent except three, But I understand the significance of the 1970s reference to give the justification for the current need of this particular study.
  3. The study aimed to identify gender differences and also a retrospective design, a specific hypothesis was not stated in the study.
  4. study have followed the scientific steps and most of the tables can be interpreted by the readers.
  5. Conclusions are made based on the study findings.
FOR  EDITORS & AUTHORS
  1. Include a statement on data analysis in the Abstract.
  2. There is no clarity regarding the assessment of physical burden . It was shown in table 3. Is it physical burden or physical comorbidities?
  3. in table 1, under physical comorbidities its mentioned as 1,2 and 3+. I understand it as number of physical comorbidities. Its better to have a foot note for better understanding by the readers.
  4. Whether permission is obtained to access the discharge records of the patients from the hospital authority?
  5. whether all the patient records were complete to extract the required data?

Author Response

  • Include a statement on data analysis in the Abstract.

Answer: Thank you for the constructive comment.  A statement was added in the abstract, as suggested.

  • There is no clarity regarding the assessment of physical burden. It was shown in table 3. Is it physical burden or physical comorbidities?

Answer:Thank you for the suggestion that allows us to clarify. In the text and in the tables we have specified that we refer to the number of physical comorbidities according to Elixhauser. The term “comorbidities” was added next to the numbers.

The term “physic burden” was replaced with “physical comorbidities” in order to make the text clearer and more understandable.

  • in table 1, under physical comorbidities its mentioned as 1,2 and 3+. I understand it as number of physical comorbidities. Its better to have a foot note for better understanding by the readers. (Fabrizio)

Answer: Thank you for your suggestion. As also described above, the term “comorbidities” was added next to the numbers. The term “physic burden” was replaced with “physical comorbidities” in order to make the text clearer and more understandable.

  • Whether permission is obtained to access the discharge records of the patients from the hospital authority?

Answer: Thank you for the suggestion that allows us to clarify. The present study was carried out in conformity with the regulations on data management of the Italian Law on privacy (Art. 20 21 DL 196/2003) published on the Official Journal n. 190 of 14 August 2004. Data were encrypted before the analysis at the Local Health Autority statistical office, where each patient was assigned a unique identifier. This identifier eliminated the possibility of tracing the patient’s identity.

  • whether all the patient records were complete to extract the required data?

Answer: Thanks for this comment. Hospital Discharge Records must pass completeness and compliance criteria to be validated and sent as a flow. Among these there are mandatory fields that must be populated for the HDR to be validated.

Reviewer 2 Report

The article by Cedrone and colleagues aim to discuss the gender differences among subjects with mental health issues in terms of Potentially Preventable Hospitalizations (PPH), an indirect indicator of existing barriers to primary care access. Understanding whether gender inequalities may play a role in accessibility to primary care could help in clarifying how modifiable and non-modifiable factors impact on healthcare pathways. Moreover, since stigma and prejudice represent relevant topics among mental health stakeholders, this research could draw attention to these problems from the public healthcare point of view, maybe leading to design preventive strategies and guiding targeted health policies.

From a formal perspective, the paper is fluid, the language is perfectly understandable, and the structure is adherent to scientific standards. Another element of significance is the large amount of data (representing an entire Local Health Authority, with more than 119,000 hospital discharge reports) collected by the Authors. The analysis process is well described and coherent in each part of the manuscript, and Results are presented quite clearly and comprehensively. The Discussion and Conclusion sections provide the reader with the tools to place the results in the right context, enabling proper understanding and providing adequate information on the subject.  

Although the paper is undoubtedly of quality, there are few minor issues I would like to bring to the attention of the Authors. 

  • In the final part of the Abstract, the lines “In males and females, alcohol abuse showed a protective role towards admissions for PQI-91 and PQI-92, …” seem to be in contrast with the Results paragraph. In fact, Odds Ratios of these two parameters reported in Tables 3 and 4 seem to have opposite direction of the effect. I would recommend clarifying this point. 
  • In the middle of the Materials and Methods paragraph, the words “performed” and “during” should be divided by a space. 
  • Considering the likelihood of multiple hospitalisations for the same subject, please indicate how multiple medical records were processed.
  • In the Table 2 there are some asterisk symbols: I would suggest adding a footnote with the corresponding caption. 
  • In Tables 3 and 4 I would suggest adding a symbol to highlight the statistically significant results. 
  • Why was the heading “physical comorbidities” changed to “physic burden” in Table 3 and Table 4? It is recommendable making the wording consistent with the other tables and with the text. Otherwise, if the purpose was to emphasize a different concept, I recommend explaining the meaning of “physic burden”.
  • In the middle of the second paragraph on page 8 (Discussion section) there is the acronym “PHC”, which is not made explicit elsewhere. I recommend to explicit it in the text before using the abbreviated form. 
  • Among the References, there are few which are not formatted like the others. In particular, reference number 12, 23, and 24 should be homologated to the others following the MDPI “Instructions for Authors”, which are strictly followed elsewhere.

Author Response

  • In the final part of the Abstract, the lines “In males and females, alcohol abuse showed a protective role towards admissions for PQI-91 and PQI-92, …” seem to be in contrast with the Results In fact, Odds Ratios of these two parameters reported in Tables 3and 4 seem to have opposite direction of the effect. I would recommend clarifying this point.

Answer: Thank you for the note. There was a typo: we have corrected it in the abstract.

  • In the middle of the Materials and Methodsparagraph, the words “performed” and “during” should be divided by a space. FATTO

Answer: Thank you for the note. We have corrected typos

  • Considering the likelihood of multiple hospitalisations for the same subject, please indicate how multiple medical records were processed.

Answer: Thanks for this valuable comment. The aim of our research was to assess the quality of primary care in a cross-sectional way. In this research we considered every preventable and non-preventable hospitalization regardless of the individuals who produced them.

  • In the Table 2there are some asterisk symbols: I would suggest adding a footnote with the corresponding caption.

Answer: Thanks for this comment. The footnote with the corresponding caption was added.

  • In Tables3 and 4 I would suggest adding a symbol to highlight the statistically significant results.

Answer: Thanks for this helpful tip. The symbol to highlight statistically significant results was added.

  • Why was the heading “physical comorbidities” changed to “physic burden” in Table 3and Table 4? It is recommendable making the wording consistent with the other tables and with the text. Otherwise, if the purpose was to emphasize a different concept, I recommend explaining the meaning of “physic burden”.

Answer: Thank you for this constructive comment which allows us to clarify. We used a variable constructed as the number of physical comorbidities according to the Elixhauser coding. We have clarified this aspect in the tables and in the text.

  • In the middle of the second paragraph on page 8 (Discussionsection) there is the acronym “PHC”, which is not made explicit elsewhere. I recommend to explicit it in the text before using the abbreviated form.

Answer: Thanks for this comment. We have replaced the acronym “PHC” with “primary health care”

  • Among the References, there are few which are not formatted like the others. In particular, reference number 12, 23, and 24 should be homologated to the others following the MDPI “Instructions for Authors”, which are strictly followed elsewhere.

Answer:Thanks for this note. We have formatted all references in such a way as to make them homogeneous following the MDPI “Instructions for Authors”.

Reviewer 3 Report

The writing of the manuscript is difficult to follow. One problem is the number of acronyms. Another one is how the authors refer to “protective role” without relating in words to the clinical situation.

In the methods the use of PQI-90, PQI-91, PQI-92 and PQI-93 is overwhelming. Please devise a way to respect the AHRQ definitions but help the reader.

For the results, some of the numbers use commas and one use a period to distinguish thousands from hundreds.

It is not clear why nonresidents, children and trauma were excluded. Please clarify in the introduction or methods.

Double diagnosis should be dual diagnosis or both diagnoses.

Table 2 is dense and hard to comprehend.

Please do not use the term “drug-abusing patients” or “alcohol abusers”. The words are stigmatizing and not allowed in top journals.

Please explain “Alcohol abuse and the presence of at least one addiction diagnosis…” Alcohol abuse is an addiction diagnosis.

What is “Physic burden”?

“Ultimate” consequence is death.

I don’t know what “numerosity” is.

Overall, it is an interesting topic but it is hard to grasp the implications of the findings as the writing obscures the points.

Author Response

  • The writing of the manuscript is difficult to follow. One problem is the number of acronyms.

Answer: Thank you for this constructive comment which allows us to improve the manuscript and its reading. We are aware of the important number of acronyms contained in the paper. Therefore, to facilitate reading, in the points where we name the PQIs, and even in the abstract, we added what kind of preventable hospitalizations they refer to. For example we tried to never mention "PQI-91" without adding "acute preventable hospitalizations" and so on.

  • Another one is how the authors refer to “protective role” without relating in words to the clinical situation.

Answer: Thank for this comment. The reference to "protective" concerns the statistical association between the variables and is not clinical at all.

  • In the methods the use of PQI-90, PQI-91, PQI-92 and PQI-93 is overwhelming. Please devise a way to respect the AHRQ definitions but help the reader.

Answer: Thank you for this constructive comment.  In methods, we have reduced the number of times PQIs are nominated

  • For the results, some of the numbers use commas and one use a period to distinguish thousands from hundreds.

Answer: Thanks for the note. The comma was added instead of the period.

  • It is not clear why nonresidents, children and trauma were excluded. Please clarify in the introduction or methods.

Answer: Thanks for making this comment. Our intention was to assess the quality of primary care in our local health authority. As suggested by the AHRQ methodology we excluded those under the age of 18; excluding users not within the competence of the LHA as primary care that treats chronicity does not depend on our service; trauma excluded because for the statistical analysis we considered it appropriate to "clean up" the non-preventable hospitalizations from violent events.

  • Double diagnosis should be dual diagnosis or both diagnoses.

Answer: Thanks for the note.  We replaced the expression “double diagnosis” with “Both diagnoses”

  • Table 2 is dense and hard to comprehend.

Answer: Thank you for this constructive comment. Table 2 was split in two tables to make it easier to read.

  • Please do not use the term “drug-abusing patients” or “alcohol abusers”. The words are stigmatizing and not allowed in top journals.

Answer: Thank you for the hint. As suggested we modified the words “abusing” in “using” and the word “abusers” with “users”, when referring to patients, as recommended by NIH article “Words Matter - Terms to Use and Avoid When Talking About Addiction”. For what concerns diagnostic categories, we kept the “abuse” term to be consistent with the ICD9-cm classification ICD-9-CM is the official system of assigning codes to diagnoses and procedures associated with hospital

  • Please explain “Alcohol abuse and the presence of at least one addiction diagnosis…” Alcohol abuse is an addiction diagnosis.

Answer: Thank you for this constructive comment. With "Alcohol abuse and the presence of at least one addiction diagnosis ..." We refer to two of the independent variables examined in our study. In particular, the "alcohol abuse" variable contains hospital discharge records with a diagnosis of alcohol abuse, while the variable "at least one addiction diagnosis" contains hospital discharge records that present either a diagnosis of alcohol abuse or a diagnosis of drug abuse. To make it easier for the reader to understand we have made that expression clearer.

  • What is “Physic burden”?

Answer: Thank you for this constructive comment. The term “physic burden” was replaced with “physical comorbidities” in order to make the text clearer. The term “comorbidities” was added next to the numbers (0, 1, 2, 3+) to make the text more understandable.

  • “Ultimate” consequence is death.

Answer: Thank you for the note. Since the expression "ultimate consequences of their health conditions" is misleading, we have replaced it with "complications from their medical conditions"

  • I don’t know what “numerosity” is.

Answer: Thank you for this comment. With the term "numerosity" we mean "numerical representativeness". For a better understanding of the text, we have replaced it.

  • Overall, it is an interesting topic but it is hard to grasp the implications of the findings as the writing obscures the points.

Answer: Thanks for making this comment. The authors believe that the changes they have made thanks to the constructive suggestions of the three reviewers have improved the reading of the manuscript.